

**Vegetation-mediated surface soil organic carbon formation and potential carbon**
**loss risks in Dongting Lake floodplain, China**
Liyan Wang[1, 2, 3], Zhengmiao Deng[1, 2], Yonghong Xie[1, 2], Tao Wang[1, 2], Feng Li[1, 2], Ye'ai Zou[1, 2],
Buqing Wang[4], Zhitao Huo[4], Cicheng Zhang[5], Changhui Peng[6], Andrew Macrae[7]
*[1] Institute of Subtropical Agriculture*, *Chinese Academy of Sciences*, *Changsha 410125, China*
*[2] Dongting Lake Station for Wetland Ecosystem Research*, *Institute of Subtropical Agriculture*,
*Chinese Academy of Sciences, Changsha 410125, China*
*[3] University of Chinese Academy of Sciences*, *Beijing 100049, China*
*[4] Changsha General Survey of Natural Resources Center, China Geological Survey, Changsha*
*410600, China*
*[5] College of Geographic Science, Hunan Normal University, Changsha 410081, China*
*[6] Department of Biological Sciences, the University of Québec at Montreal, Montreal, QC H3C 3P8,*
*Canada*
*[7] Centro de Ciências da Saúde (CCS), Universidade Federal do Rio de Janeiro, BR 21941902, Brazil*
**Corresponding author:** Zhengmiao Deng (dengzhengmiao@163.com) and
Yonghong Xie (yonghongxie@163.com)
**Abstract**
Sources and stabilization mechanisms of soil organic carbon (SOC) fundamentally
govern the carbon sequestration potential of wetland ecosystems. Nevertheless,
systematic investigations regarding SOC sources and molecular stability remain scarce
in floodplain wetland environments. This study employed dual analytical approaches
(stable isotope analysis and $^{13}$C nuclear magnetic resonance spectroscopy) to
characterize surface SOC composition across three dominant vegetation communities
(*Miscanthus*, *Carex*, and mudflat) in Dongting Lake floodplain wetlands. Key findings
revealed: (1) Significantly elevated SOC concentrations in vegetated communities
(*Miscanthus*: 13.76 g kg$^{-1}$; *Carex*: 12.98 g kg$^{-1}$) compared to unvegetated mudflat (6.88
g/kg); (2) Distinct $\delta^{13}$C signatures across communities, with the highest isotopic values
in *Miscanthus* (-22.67 ‰), intermediate in mudflat (-26.01 ‰), and most depleted
values in *Carex* (-28.25 ‰); (3) Bayesian mixing models identified autochthonous plant



biomass as the primary SOC source (*Miscanthus*:53.3±10.6 %, *Carex*:52.4 %±11.6 %,

Mudflat:47.5±12.5 %); (4) Spatial heterogeneity in POM contributions across sub-

lakes, showing descending contributions from South (highest) > West > East (lowest)

Dongting Lake; (5) Molecular characterization revealed O-alkyl C dominance (27.3-

46.8 %), followed by alkyl C and aromatic C. Notably, *Miscanthus* soils exhibited

enhanced O-alkyl C content (Alip/Arom) and reduced aromaticity/hydrophobicity

indices, suggesting comparatively lower biochemical stability of its SOC pool. These

results highlight the critical role of vegetation-mediated SOC formation processes and

warn against potential carbon loss risks in *Miscanthus*-dominated floodplain

ecosystems, providing a scientific basis for carbon management of wetland soils.

**Keywords:** Floodplain wetland; Stable isotope; Soil carbon source; $^{13}$C NMR; Organic

carbon stability

**1 Introduction**

  Although wetlands occupy merely 5-8 % of the global terrestrial surface, they

disproportionately store 20-30 % of the terrestrial carbon, positioning them as pivotal

regulators in global carbon cycling (Kayranli et al., 2010; Köchy et al., 2015; Mitsch et

al., 2013). Small changes in wetland soil organic carbon (SOC) stocks may have large

feedback effects on climate-carbon cycle interactions. The long-term carbon

sequestration capacity of wetland ecosystems is jointly governed by two critical factors:

carbon input dynamics and biochemical stabilization mechanisms. Therefore, clarifying

the sources and stabilization pathways of wetland SOC is essential for optimizing

carbon sink management and enhancing climate change mitigation strategies.

  In floodplain systems, the organic carbon in sediment derives from both

autochthonous (in-situ plant biomass and aquatic plankton) and allochthonous sources

(river-transported particulate organic matter, POM) (Robertson et al., 1999). Notably,

the relative contributions of these sources vary significantly across vegetation

communities, driven primarily by vegetation characteristics (e.g., biomass production

and litter composition) and hydrological regimes (e.g., flood duration, frequency, and

intensity). For example, in mangrove ecosystems, mangrove community SOC is mainly



derived from mangrove plant tissues, whereas adjacent *S. alterniflora* and tidal flats
exhibit stronger reliance on fluvially imported POM (Wang et al., 2024a). These source
differences across vegetation communities are further modulated by geomorphic
features (e.g., elevation, channel morphology) and anthropogenic disturbances (e.g.,
land-use changes). For example, topographic complexity determines the lateral
transport and deposition of POM in rivers, resulting in localized heterogenic carbon
accumulation. Despite these insights, critical knowledge gaps persist regarding
interspecific differences in carbon sourcing among co-occurring vegetation
communities within floodplain wetlands and the spatial scaling of these heterogeneities.
Stable carbon and nitrogen isotopes have been widely used to analyze the sources of
wetland SOC (Sasmito et al., 2020; Wu et al., 2021a).
SOC stability is defined as the capacity of organic compounds to
resist changes and/or losses (Doetterl et al., 2016). Enhanced SOC stability typically
corresponds with preferential accumulation of recalcitrant compounds that withstand
microbial degradation. $^{13}$C nuclear magnetic resonance (NMR) is widely used to
analyze the chemical composition of SOC, and can calculate the relative abundance of
various C functional groups closely related to SOC decomposition (Shen et al., 2018).
Biochemically recalcitrant components include alkyl-C and aromatic-C, whereas labile
components comprise O-alkyl-C and carbonyl-C (Skjemstad et al., 1994) .
Consequently, soils enriched in labile SOC fractions demonstrate heightened
vulnerability to carbon loss through accelerated decomposition pathways, particularly
under environmental disturbance. These molecular signatures are regulated by fators,
including vegetation inputs (via lignin/cellulose ratios and aliphatic content), soil
properties (clay-silt particle associations), and climatic controls on vegetation litter
decomposition (Cano et al., 2002; Chen et al., 2018; Liu et al., 2022; Preston et al.,
1994; Quideau et al., 2001; Wu et al., 2020). In floodplain environments, hydrologic
conditions further regulate SOC components by affecting oxygen supply and altering
microbial metabolism and enzyme activity (Kirk and Farrell, 1987; Boye et al., 2017).
However, there are insufficient studies on the sources and stability of SOC in floodplain





wetlands.

Dongting Lake, a Yangtze River-connected floodplain wetland, presents an ideal

natural laboratory for investigating these processes. Its elevation-dependent vegetation
zonation and complex topography create pronounced gradients in carbon source inputs
and stabilization conditions. Among soil carbon pools, surface SOC is more susceptible
to the effects of climate, hydrological conditions and human activities, resulting in a
high carbon turnover rate and requiring more attention. In this study, stable isotope
techniques were used to analyze the source of surface SOC and the stability of SOC
was further evaluated using the $^{13}$C NMR method. The hypotheses of this study were
(1) with regard to vegetation communities, based on plant biomass, SOC content should
be the highest in the *Miscanthus* community, followed by the *Carex* community, with
the Mudflat exhibiting the lowest SOC content. From a spatial perspective, considering
the influence of topographic and hydrological characteristics, the SOC levels were
expected to follow a gradient, being highest in East Dongting Lake, intermediate in
South Dongting Lake, and lowest in West Dongting Lake. (2) SOC in *Miscanthus* and
*Carex* community would primarily originate from autochthonous plant sources due to
their high biomass production; in contrast, the source of SOC in the Mudflat would
primarily originate from allochthonous POM, and (3) due to the difference in sources,
the SOC structure in *Miscanthus* and *Carex* should be dominated by O-alkyl C, and the
SOC structure of the Mudflat should be dominated by aromatic C.

**2 Materials and methods**
**2.1 Study areas**
Dongting Lake (28°30′–30°20′N, 111°40′–113°10′E) is the second largest inland
freshwater lake in China, with an area of 2564 km$^2$. It comprises East Dongting Lake
(EDL, 1327.8 km²), West Dongting lake (WDL, 443.9 km²) and South Dongting Lake
(SDL, 920 km²) (Jun-Feng et al., 2001). The Lake is a typical river-connected lake that
mainly receives inflow from the Yangtze River through three channels (the Songzi,
Hudu, and Ouchi Rivers) and other four tributaries (the Xiang, Zi, Yuan, and Li Rivers)





and then outflows into the Yangtze River from the Chenglingji outlet (Deng et al., 2018).
The lake's water level exhibits significant seasonal fluctuations, with flood periods
occurring from June to October. From the water's edge to the uplands, the dominant
vegetation communities include Mudflat communities, *Carex spp.* (Cyperaceae)
communities, and *Miscanthus sacchariflorus* (Poaceae) communities (Xie et al.,
2015).The study area is characterized by a humid subtropical monsoon climate with a
mean annual temperature of 16.8°C and a mean annual precipitation of 1382 mm.

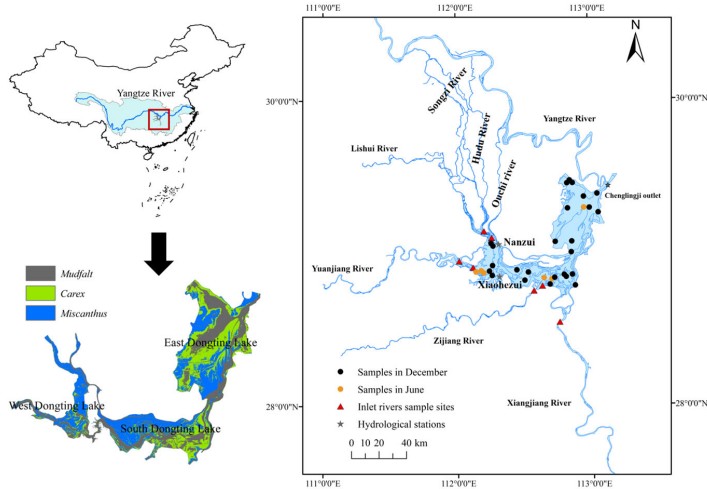


**Figure 1.** Map of the study area and sampling sites (base map from ESRI).
**2.2 Field sampling and parameter measurement**
Soil sampling was conducted across three dominant vegetation during December
2022, with supplementary Mudflat sediment sampling in June to account for
hydrological accessibility constraints. The final sampling comprised 31 sampling sites
(11 Mudflat, 8 *Carex* community, 12 *Miscanthu*s community) with latitude and
longitude recorded using a hand-held global positioning system (GPS). Notably, *Carex*
communities in West Dongting Lake were excluded from sampling due to insufficient
population density. At each sampling site, a 1x1 m sample plot was set up, and surface
(0-20 cm, 500 g fresh soil) soil samples were collected from five points in the plot and
mixed for subsequent analysis. For vegetated sites (*Carex* and *Miscanthus*
communities), aboveground tissue, surface litter layer and belowground roots were





collected from the sample plots. All samples were transported to the laboratory. Soil
samples were air-dried in a cool, ventilated area, ground and passed through a 0.147
mm sieve for subsequent analysis. Plant material was dried at 60°C to a constant mass
and the dry weight was recorded prior to pulverization. Both SOC and plant organic
carbon content was quantified using the potassium dichromate-sulfuric acid oxidation
technique. The TN content of soil was measured using an elemental analyzer (Vario
MAX CNS, Elementar, Germany). The formula for calculating vegetation organic
carbon stocks (VOCS) is as follows:
$VOCS = A \times VB \times VOC$  (1)
Where A is the vegetation distribution area (km$^2$), VB is the vegetation biomass
(t/km$^2$), VOC is the vegetation organic carbon content (g kg$^{-1}$).

## 2.3 Inundation duration and runoff volume

We used the hydrological data from Chenglingji, Xiaohezui, and Nanzui
hydrological stations to calculate the inundation time and runoff volume of EDL, SDL,
and WDL, respectively. The hydrological data from Chenglingji, Xiaohezui and Nanzui
have been widely used to analyze the hydrological characteristics of EDL, SDL and
WDL. Vegetation is classified as submerged when water levels exceed specific
elevations. Using daily water levels and elevation data from the Dongting Lake Wetland
DEM (Geospatial Data Cloud: http://www.gscloud.cn), we calculated vegetation-
specific inundation durations. The formula for calculating the inundation duration is as
follows:
The inundation duration (ID) was calculated as follows:
$ID=\sum_{WD>0}^{n} I_{WD}$  (2)
$WD=WL-E$  (3)
where WL is the water level at the Chenglingji (EDL), Xiaohezui (SDL), and Nanzui
(WDL) Hydrological Station, E is the elevation, $I_{WD}$ is the number of days when WD>0,
and n is the number of days per year.

## 2.4 Stable isotope analysis and mixing model

The soil samples (2 g) were added to 0.5 mol/L hydrochloric acid reflections for



24 h to removal carbonates, then washed to neutrality with distilled water and dried at
55 °C. The treated soil samples were ground through a 0.147 mm sieve and used for
stable isotope measurements. $\delta^{13}$C and $\delta^{15}$N stable isotope ratios were measured using
a gas chromatography-isotope ratio mass spectrometer (Delta V advantage, Thermo
Fisher) and were calculated from the following equation:
$$\delta(‰) = \left( \left( R_{sample} / R_{standard} \right) - 1 \right) \times 1000 \qquad (4)$$
where $R_{sample}$ is the stable $^{13}$C/$^{12}$C or $^{15}$N/$^{14}$N isotope ratio of the sample, and $R_{standard}$ is
stable the $^{13}$C/$^{12}$C or $^{15}$N/$^{14}$N isotope ratios of the international isotope standard (Vienna
Peedee Belemnite and $N_2$ in the atmosphere, respectively).
SOC potential sources include *Miscanthus* plant, *Carex* plant and Plankton, and
rivers suspended particulate organic matter (POM). In addition to plankton, we
collected other potential end-members for stable isotope analysis. Five samples of
aboveground tissues, surface litter and root of *Miscanthus* and *Carex* plants were
randomly sampled. Due to the construction of the Three Gorges Dam, the POM entering
Dongting Lake changed from three channels (the Songzi, Hudu, and Ouchi Rivers) to
four tributaries (the Xiang, Zi, Yuan, and Li Rivers) (Wang et al., 2024b). Therefore,
we collected POM at the inlets of the Xiang, Zi, Yuan, and Li Rivers into the lake. The
POM from the Yuan and Li Rivers served as the allochthonous end-members for WDL,
while the POM from the Xiang, Zi, Yuan, and Li Rivers served as the allochthonous
end-members for EDL and SDL (Fig. 1).
Source contributions were quantified using a Bayesian mixing model based on
$\delta^{13}$C and $\delta^{15}$N. The MixSIAR model combines the advantages of SIAR and MixSIR. It
not only introduces fixed and random effects, but also incorporates source uncertainty.
These features endow the MixSIAR model with higher source analysis accuracy, and it
has been widely used in wetland sediments (Zhang et al., 2024).
**2.5 $^{13}$C NMR analysis and spectral indices**
The chemical structure of SOC was determined by solid-state $^{13}$C NMR
spectroscopy. In order to improve the signal-to-noise ratio, soil samples are pretreated
with hydrofluoric acid (HF) before $^{13}$C NMR spectroscopy analysis. Soil samples (8.0





g) were placed into 100 mL plastic centrifuge tubes containing 50 mL of 10% (v/v) HF
solution. The tubes were shaken on a shaking bed at 200 rpm for 1 hour at 25 °C, then
centrifuged at 3800 rpm for 5 minutes. After discarding the supernatant, the residual
soil was subjected to repeated HF treatments under identical conditions. The entire
procedure was conducted 8 times with the following shaking durations: 1 hour for the
first 4 cycles, 12 hours for cycles 5-7, and 24 hours for the final cycle. The treated
residue was washed 5-6 times with distilled water to remove the HF solution. The
residue was dried in an oven at 40 °C and sieved through 0.25 mm sieve. Subsequently,
pretreated samples were analyzed using a Bruker AVANCE III HD 600MHz
spectrometer equipped with an H/X dual-resonance solid probe, operating in CP/MAS
mode. Experimental parameters were set as follows: 4-mm $ZrO_2$ rotor spinning at 10
kHz, $^{13}C$ detection resonance frequency of 150 MHz, acquisition time of 6.25 μs, and
spectral width of 30 kHz.
The spectra of samples were divided in the following chemical shift regions: 0–45
ppm (alkyl C, originating from Microbial metabolites and plant biopolymers), 45–110
ppm (O-alkyl C, derived from carbohydrates), 110–160 ppm (aromatic C, derived from
lignin, polypeptides and black carbon) and 160–220 ppm (carbonyl C, derived from
fatty acids, amino acids and lipids). The relative abundances of different carbon
functional groups were quantitatively determined by integrating their respective peak
areas in the solid-state $^{13}C$ NMR spectra. Subsequent spectral analyses were performed
using MestReNova software (12.0.0-20080) for statistical interpretation of the data.
SOC spectra of the different communities are provided in the Appendix A (Fig. S1).
According to (Boeni et al., 2014; Wang et al., 2023), four indicators of the stability of
SOC were calculated as:
(1) A/O-A, which is used to indicate the degree of humification of SOC, the higher
the value, the more resistant it is to decomposition
(2) Alip/Arom, which is used to indicate the complexity of the molecular structure
of humus, the higher the ratio, the simpler the molecular structure;
Alip/Arom = (alkyl C+ O-alkyl C)/aromatic C



(3) aromaticity index (AI), which is used as measure of the complexity of SOC
structure;
AI = aromatic C/ (alkyl C+ O-alkyl C+ aromatic C)
(4) hydrophobicity index (HI), which is used to indicate the stability of SOC
integrated with aggregates.
HI = (alkyl C+ aromatic C)/ (O-alkyl C+ carbony C)
**2.6 Statistical analysis**
The Shapiro-Wilk test and the Levene test are used respectively to test the
regularity and consistency of the data. Differences between community were evaluated
through one-way analysis of variance (ANOVA); multiple comparisons were performed
using the least significant difference (LSD) test.  Nonparametric tests were used for data
that did not meet homogeneity of variance. A threshold of $P<0.05$ was used to denote
statistically significant differences. Source contributions were quantified using the
"MixSIAR" package in R.

**3 Results**
**3.1 Hydrological Characteristics of East, South and West Dongting Lakes**





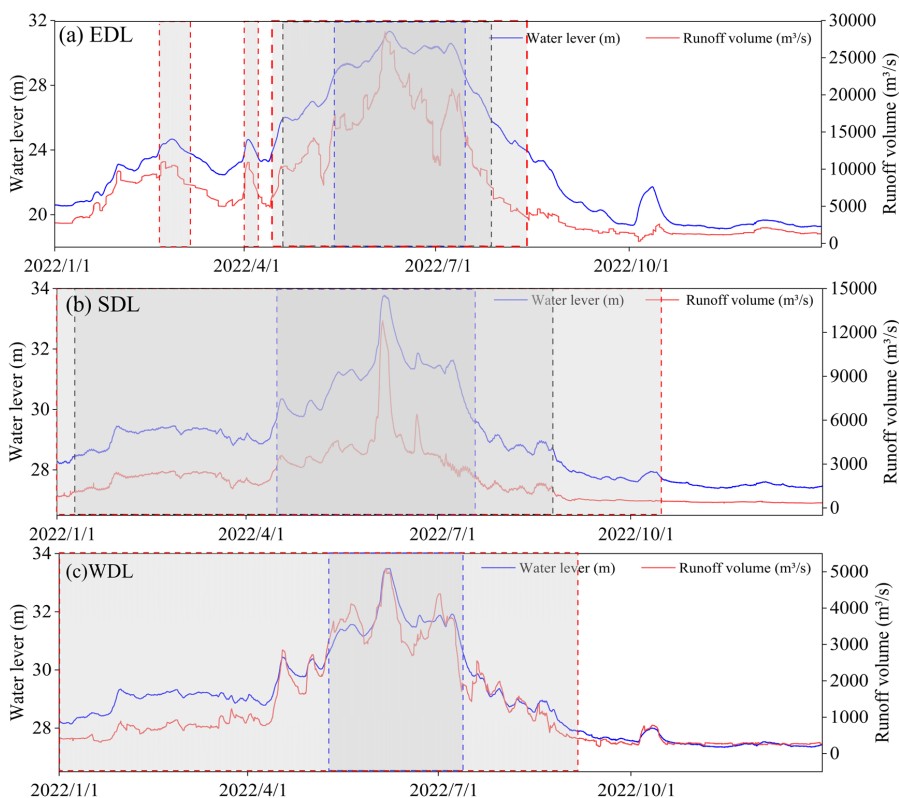

**Figure 2.** Water level, Runoff volume and inundation duration in EDL, SDL, and WDL. In the figure, the shaded part represents submerged, with the red, black, and blue dashed boxes respectively indicating the Mudflat, *Carex*, and *Miscanthus* communities. EDL: East Dongting Lake; SDL: South Dongting Lake; WDL: West Dongting Lake.

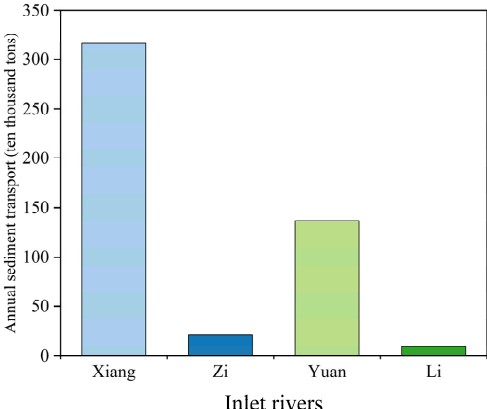

**Figure 3.** The annual sediment transport of inlet rivers in 2022.



The water level of Dongting Lake shows significant fluctuations (19.24-33.78 m) (Fig.2). There were differences in the inundation duration of different vegetation communities, with the Mudflat having the longest inundation duration (223.8 d), followed by *Carex* (162.4 d), and *Miscanthus* having the shortest inundation time (78.9 d). Among the sub-lakes, SDL showed the longest inundation time (206.8 d), followed by WDL (152 d) and EDL (102.8 d). The annual runoff volume was the highest in EDL, followed by SDL and WDL. The annual sediment transport of four tributaries was 484.1 ×10$^4$ tons, with the Xiangjiang River having the highest annual sand transport (Fig.3).

**3.2 Carbon sink capacity in dominant vegetation community**

The area of Dongting Lake wetland spans 2564.1 km², with vegetation distribution dominated by the *Miscanthus* community (36.9 %), followed by the Mudflat (33.0 %) and the *Carex* community (30.1 %) (Table 1). *Miscanthus* community exhibited significantly higher plant biomass (2922.9 t/km$^2$) and tissue carbon content (454.7 g kg$^{-1}$) than *Carex* community (1391.0 t/km$^2$ and 422.4 g kg$^{-1}$, respectively; $P < 0.05$). Consequently, its organic carbon stock (1.258 ± 0.13 Tg C) nearly tripled that of *Carex* communities, representing 72.5 % of the wetland's total vegetation-mediated carbon storage.

**Table 1**

Distribution area, biomass, organic carbon content and carbon stock in dominant vegetation community.

| Community types | Areas(km$^2$) | vegetation biomass (t/km$^2$) | vegetation organic carbon content (g kg$^{-1}$) | vegetation organic carbon storage (Tg C) |
|---|---|---|---|---|
| *Miscanthus* | 946.74 | 2922.9±300.8a | 454.7±6.22a | 1.258±0.13a |
| *Carex* | 770.63 | 1391.0±269.7b | 422.4±4.75b | 0.453±0.09b |
| *Mudflat* | 846.72 | 0 | 0 | 0 |

**3.3 Stable isotope of soil and vegetation**

*Miscanthus* plants displayed the most enriched δ$^{13}$C values (-13.85 ‰ to -17.24 ‰), contrasting with plankton-derived carbon showing the most depleted signatures. Conversely, δ$^{15}$N values followed an inverse pattern, with plankton exhibiting the highest enrichment (Table 2). There were differences in SOC and TN





contents among community types, with the *Miscanthus* and *Carex* communities having
significantly higher SOC and TN contents than the Mudflat community ($P < 0.05$, Fig.
4a).
The soil $\delta^{13}C$ value ranged from -30.85 to -18.01‰ (-25.30±0.54 ‰) with the
highest values were observed in *Miscanthus* (-18.01 to -26.08 ‰) ($P < 0.05$, Fig. 4a),
followed by *Mudflat* (-24.3 to –28.68 ‰) and *Carex* (-27.08 to -30.85 ‰). There was
no significant difference in soils $\delta^{15}N$ values from different vegetation types. EDL
*Carex* communities were smaller in $\delta^{13}C$ compared to SDL ($P < 0.05$, Fig. 4b), while
other vegetation types showed no significant inter-regional differences in SOC, TN,
$\delta^{13}C$ or $\delta^{15}N$ across sub-basins (Fig. 4b).
**Table 2**
Carbon and nitrogen stable isotope signatures (‰) of different potential end-members

| Sources | $\delta^{13}C$ (‰) | $\delta^{15}N$ (‰) | |
|---|---|---|---|
| *Miscanthus* Plant | -14.46±0.63 | 0.2±1.45 | 288 |
| Carex Plant | -29.51±0.27 | 2.42±1.03 | 289 |
| EDL+SDL POM | -29.31±1.08 | 6.38±1.5 | 290 |
| WDL POM | -29.22±1.40 | 6.08±1.82 | 291 |
| Plankton* | -30.0 ±6.60 | 6.5±0.75 | 292 |
| | | | 293 |

* C and N stable isotope signature of Plankton were cited from (Kendall et al., 2001;
Li et al., 2016)

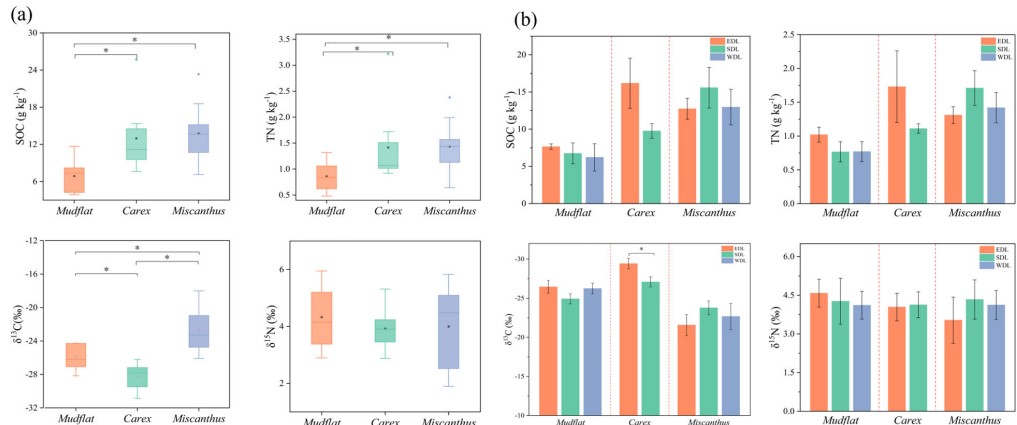


**Figure 4.** Characteristics of SOC, TN, $\delta^{13}C$ and $\delta^{15}N$ with vegetation types (a), and in
different sub lakes(b). EDL: East Dongting Lake; SDL: South Dongting Lake; WDL:



West Dongting Lake.

**3.4 SOC sources and contribution**

The isotopic composition of all soil samples fell within the mixing space delineated by potential end-members, confirming their effectiveness in source discrimination (Fig. 5). Our study showed autochthonous plant (including *Miscanthus* and *Carex* plant) was the main source of SOC in Dongting floodplain wetland (*Miscanthus*:53.3±10.6 %, *Carex*:52.4%±11.6 %, Mudflat:47.5±12.5 %)(Fig. 6a). Allochthonous POM contributions exhibited significant variation across vegetation types, with minimum values in *Miscanthus* communities (26.8 ± 8.1 %) versus *Carex* (31.3 ± 8.3 %) and mudflat (35.4± 10.2 %).

Spatial heterogeneity in carbon source contributions was evident across vegetation types (Fig. 6b). In *Miscanthus* communities, EDL demonstrated maximal autochthonous input dominance (12.1% and 13.9% greater than SDL and WDL respectively), whereas allochthonous POM displayed inverse spatial patterns (10.9% and 4.7% lower than SDL and WDL respectively). In *Carex* communities, EDL showed 8.1% higher in autochthonous contributions relative to SDL, concomitant with 9.1% reduce in POM inputs compared to SDL.

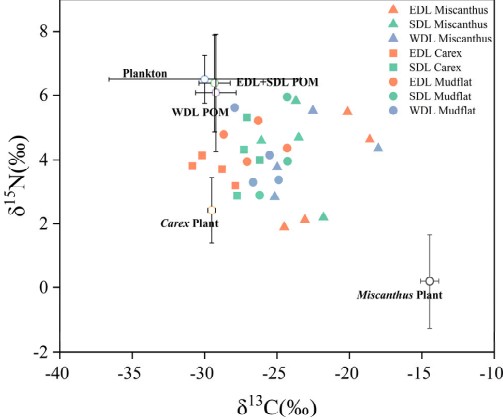

**Figure 5.** The end-element plots of $\delta^{13}$C and $\delta^{15}$N values for samples of Dongting Lake soil and SOC sources. EDL: East Dongting Lake; SDL: South Dongting Lake; WDL: West Dongting Lake.



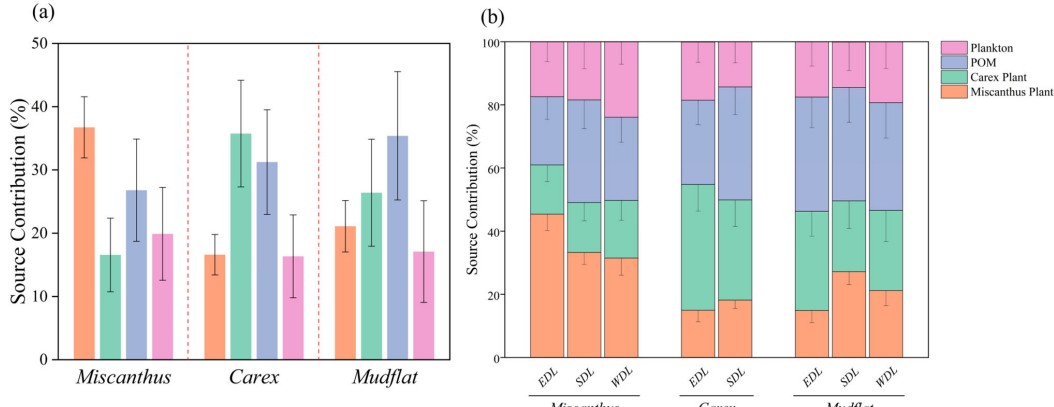

**Figure 6.** Relative contributions of SOC sources with vegetation types (a) and in different sub lakes (b). POM: particulate organic matter; EDL: East Dongting Lake; SDL: South Dongting Lake; WDL: West Dongting Lake.

**3.5 Chemical structure and SOC stability**

SOC functional groups were dominated by O-alkyl C ( 27.3–46.8 %), followed by alkyl C (17.8–41.7 %) and aromatic C (15.5–26.6 %), with Carbonyl C exhibiting minimal abundance. The highest abundance of alkyl C was observed in Mudflat community (25.4 ± 1.2 %), followed by *Carex* (23.2 ± 0.9 %), and then *Miscanthus* community (22.1 ± 0.6 %) ($P < 0.05$, Fig. 7a); O-alkyl C shows the opposite trend. The abundances of aromatic C were significantly higher in the *Carex* community than *Miscanthus* (Fig. 7a, $P < 0.05$). Carbonyl C showed the same trend as alkyl C. There were no significant changes in the abundance of SOC functional groups across vegetation types in different sub lakes (Fig. 7b).

Stability indices showed that Mudflat and *Carex* communities had significantly higher A/O-A ratios, HI indices and aromaticity than *Miscanthus* ($P < 0.05$), while the Alip/Arom ratio showed the opposite pattern (Fig. 8), suggesting that the Mudflat and *Carex* community formed a more stable organic carbon pool through enrichment of difficult-to-degrade fractions, such as alkyl C and aromatic C.





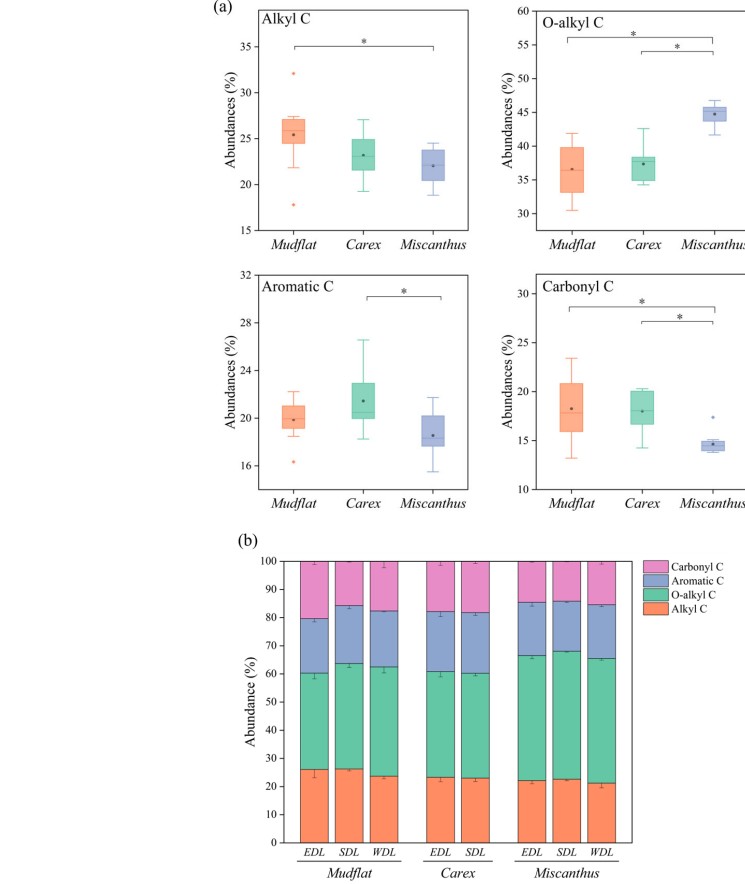

**Figure 7.** SOC functional group abundance in different vegetation types (a) and in different sub lakes (b). EDL: East Dongting Lake; SDL: South Dongting Lake; WDL: West Dongting Lake.



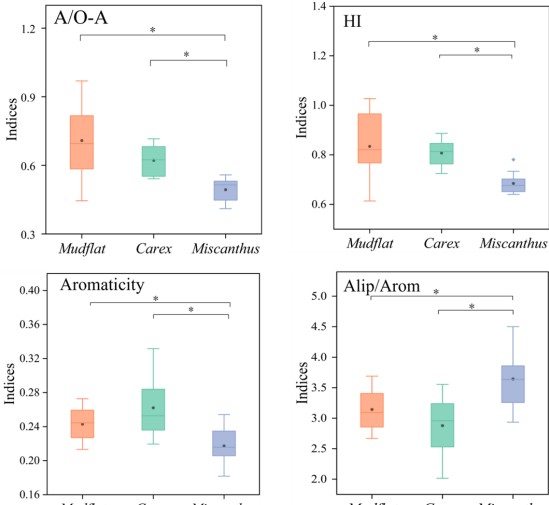

**Figure 8.** SOC stability index for different vegetation types. A/O-A: the ratio of alkyl C over O-alkyl C; HI: hydrophobicity index, the ratio of the sum of alkyl and aromatic C over the sum of O-alkyl and carbony C; Alip/Arom, the ratio of the sum of alkyl C and O-alkyl C over aromatic C; AI, aromaticity index, the ratio of aromatic C over the sum of alkyl C, O-alkyl C and aromatic C.

## 4 Discussion

### 4.1 SOC content in different vegetation types

Our study showed that the SOC content of Mudflat community (6.88 g kg$^{-1}$) was the lowest, and there was no significant difference in SOC content between the two communities (*Miscanthus*: 13.76 g kg$^{-1}$ and *Carex*: 12.99 g kg$^{-1}$). These results partially support our first hypothesis that SOC content should be the highest in the *Miscanthus* community, followed by the *Carex* community, with the Mudflat exhibiting the lowest SOC content. Although the vegetation biomass of *Miscanthus* community (2922.9 ± 300.8 t/km²) was significantly higher than that of *Carex* community (1391.0±269.7 t/km²), the simpler chemical structure of *Miscanthus* SOC (Fig.7) may facilitate its microbial decomposition. The cross-sub-lake comparisons revealed no significant spatial heterogeneity in vegetated SOC content, which was also inconsistent to our first hypothesis. This may be due to the joint influence of vegetation, hydrology and human



disturbance on SOC content.
**4.2 SOC sources in different vegetation types**
Our results showed that autochthonous plant were the main source of SOC
(*Miscanthus*:53.3±10.6%, *Carex*:52.4%±11.6%, Mudflat:47.5±12.5 %) ,which
partially supports our second hypothesis that SOC in *Miscanthus* and *Carex* community
would primarily originate from autochthonous plant sources; the source of SOC in the
Mudflat would primarily originate from allochthonous POM. The SOC of *Miscanthus*
and *Carex* communities is mainly derived from autochthonous plant which were related
to the plant biomass of communities (*Miscanthus*: 2922.9 ± 300.8 t/km², *Carex*:
1391.0±269.7 t/km²) (Table 1). Each year autochthonous plants input a large source of
carbon into the soil (Zhu et al., 2022). SOC in the mudflat community was also
predominantly derived from autochthonous plants, which can be attributed to reduced
allochthonous POM inputs. The commissioning of the Three Gorges Dam in 2003, the
world's largest hydropower project, fundamentally altered sediment dynamics, reducing
downstream sediment transport from $120 \times 10^6$ tons/year (pre-dam) to a state of net
erosion ($2 \times 10^6$ tons/year post-dam) (Yu et al., 2018). The reductions in river sediment
transport diminished allochthonous POM contributions. Autochthonous plants are also
a major source of SOC in Poyang Lake (located in the lower reaches of the Yangtze
River), riverine wetlands along Mexico's Pacific coast, and coastal wetlands in the
Mississippi River delta (Wang et al., 2016; Kelsall et al., 2023; Adame and Fry, 2016).
The source of SOC in Dongting floodplain wetland has a part of the source of plankton
(14.3-23.9 %). This is due to the decline in water quality of the lakes and the gradual
increase in algae as a result of problems such as the increased intensity of agricultural
farming and the use of chemical fertilizers (Ren et al., 2018).
POM had the highest SOC contribution to the Mudflat community (35.4±
10.2 %), followed by *Carex* (31.3± 8.3 %), and the lowest was *Miscanthus* (26.8 ±
8.1 %). This may be related to the different elevations of the vegetation communities
(*Miscanthus*:>25 m, *Carex*:22-25 m, Mudflat:<22 m), where higher elevations lead
to shorter inundation times, thus limiting particulate organic matter (POM) deposition.



In this study, we also found that SDL exhibited the highest POM contribution (32.5 %),
followed by WDL (26.3 %), with EDL showing minimal inputs (21.6 %) in *Miscanthus*
communities. A parallel pattern emerged with *Carex* communities, where SDL's POM
contribution exceeded EDL by 9.1%. This may be due to the following: Firstly, the
intensive agricultural activities and urbanization in the Xiangjiang River basins that
have increased soil erosion, making more POM enter the SDL (Xiao et al., 2023).
Second, the northern part of the SDL receives a large amount of sediment under the top-
supporting effect of the outflow of WDL (Zhang et al., 2019). Third, the inundation
duration is the longest in the SDL, followed by the WDL, and the EDL has the shortest
inundation duration. The extension of inundation duration can improve the deposition
of allochthonous POM (Shen et al., 2020). Studies have also shown that the mean mass
accumulative rate (MAR) of the SDL is the highest, followed by the WDL, and the EDL
is the lowest (Ran et al., 2023). Thus, the spatial heterogeneity of allochthonous POM
contributions to SOC across sub-lakes revealed synergistic controls by anthropogenic
and hydrodynamic drivers.
**4.3 SOC stability in in different vegetation types**
Our findings demonstrate that O-alkyl C, primarily derived from carbohydrates,
constitutes the dominant fraction (27.3 – 46.8 %) of SOC in Dongting Lake wetlands.
This result partially supports our third hypothesis that the structure of SOC in
*Miscanthus* and *Carex* should be dominated by O-alkyl C, and the SOC structure of the
Mudflat should be dominated by aromatic C. The predominance of O-alkyl C across
vegetation communities likely reflects the autochthonous origin of SOC from plant-
derived inputs. Specifically, the cellulose and hemicellulose components of plant litter
decompose rapidly to produce carbohydrates (Mckee et al., 2016),which is consistent
with findings from other lake or river wetlands where O-alkyl C represents the principal
SOC fraction (Yang et al., 2023; Wang et al., 2011)
Notably,  the *Miscanthus* community  exhibited significantly  higher O-alkyl  C
content compared to *Carex* and mudflat, while displaying lower alkyl and aromatic C
contents (Fig. 7a).  Given that O-alkyl C was classified as labile C whereas alkyl and





aromatic C were classified as recalcitrant C, these results showed that *Miscanthus*
community SOC is more unstable and more susceptible to decomposition. Therefore,
the risk of SOC loss is higher in the *Miscanthus* community. The A/O-A and aromaticity
as well as HI and Alip/Arom, are recognized as important parameters for evaluating the
stability of SOC. The A/O-A ratio, aromaticity and hydrophobicity index (HI) were
significantly higher in the *Carex* and mudflat communities than *Miscanthus* community
($P < 0.05$), whereas the Alip/Arom ratio showed the opposite trend, indicating that  the
SOC of  *Carex* and mudflat communities had more complex structures and higher
hydrophobicity, which increased SOC stability (Spaccini et al., 2006).
O-alkyl C is primarily derived from carbohydrates. *Miscanthus* plants possess a
well-developed underground root system that may produces more root secretions,
which are mainly composed of carbohydrates (Wu et al., 2021b). The higher aromatic
and alkyl C fractions observed in *Carex* and mudflat communities likely result from
prolonged inundation duration, which extends exposure to anaerobic conditions.
Anoxic conditions significantly limit reactive oxygen species generation and catalase
activity, thereby inhibiting oxidative decomposition of lignin (the main component of
aromatic carbon) (Benner et al., 1984; Kirk and Farrell, 1987). Additionally, microbial
metabolic efficiency declines under oxygen deprivation, retarding the decomposition of
lipids and waxes (alkyl carbon precursors) (Keiluweit et al., 2017). These stability
difference may be related to the contribution of allochthonous POM. Allochthonous
carbon is rich in aromatic and hydrophobic components, exhibiting stronger resistance
to decomposition (Keil, 2011). The proportion of allochthonous POM was significantly
higher in the *Carex* and mudflat communities than in the *Miscanthus*.
The risk of loss of soil carbon pools in *Miscanthus* community is higher due to
the more labile molecular structure of SOC (Fig. 9). In our previous research, we also
found that  the *Miscanthus* community experienced the greatest loss of SOC from 2013
to 2022 (Wang et al., 2025). Although the SOC stability of the *Miscanthus* community
is relatively low, its SOC content shows no significant difference from that of the *Carex*
community due to high litter input ($1.258 \pm 0.13$ Tg C), revealing the differences in the





mechanisms of carbon sequestration function formation among different vegetation
types in floodplain wetlands (Fig. 9).

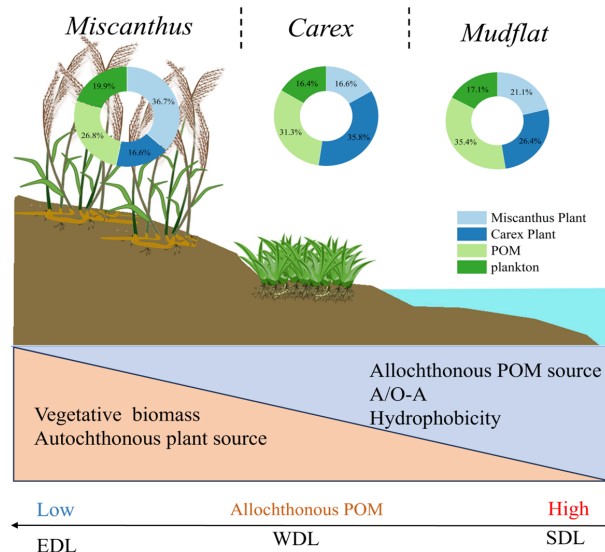


**Figure 9.** A conceptual map of the sources and stability of SOC on a geomorphic

gradient in the Dongting floodplain wetlands. Orange triangles show the decrease in
vegetative biomass and autochthonous plant sources from *Miscanthus* (high elevation)
to Mudflat (low elevation). In contrast, blue triangles show increases in allochthonous
POM sources, A/O-A, and hydrophobicity. The arrows below indicate that from SDL
to WDL to EDL, the contribution of allochthonous POM is decreasing. A/O-A: the ratio
of alkyl C over O-alkyl C; POM: particulate organic matter; EDL: East Dongting Lake;
SDL: South Dongting Lake; WDL: West Dongting Lake.


**5 Conclusions**
Stable isotopic analysis demonstrates that SOC in Dongting floodplain wetlands
was mainly derived from autochthonous plant inputs, with mean contributions of 53.3
± 10.6 % (*Miscanthus*), 52.4 ± 11.6 % (*Carex*), and 47.5 ± 12.5 % (mudflat). Notably,
allochthonous POM contributions exhibited both vegetation-dependent (mudflat >
*Carex* > *Miscanthus*) and regional disparities (SDL>WDL>EDL). We attribute these



differences to interacting effects of anthropogenic and hydrodynamic drivers, which
collectively regulate allochthonous POM transport and deposition. The A/O-A ratios,
aromaticity, and hydrophobicity were lower in *Miscanthus* community, indicating that
SOC is more easily decomposed, and the stability of SOC pools is lower. Therefore, we
should prioritize the conservation of *Miscanthus* communities SOC to mitigate carbon
loss risks.

**Acknowledgments**
This work was supported by the National Natural Science Foundation of China
(U2444221, U22A20570, U21A2009) and the Natural Science Foundation of Hunan
Province(2025JJ20039).

**Author contributions**
LW: Writing – original draft, Investigation, Data curation. ZD: Writing – review &
editing, Project administration, Funding acquisition, Conceptualization. YX: Writing–
review & editing, Funding acquisition. TW: Investigation, Data curation. FL: Writing–
review & editing, Methodology. YZ: Investigation, Data curation. BW: Formal analysis,
Resources. ZH: Methodology, Data curation. CZ: Investigation, Data curation. CP:
Writing – review & editing, Formal analysis. AM: Formal analysis, Conceptualization.

**Data availability**
Data will be made available upon request.

**Declaration of Competing Interest**
The authors declare that they have no known competing financial interests or personal
relationships that could have appeared to influence the work reported in this paper.



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
