# Peer review of "Vegetation-mediated surface soil organic carbon formation and potential carbon loss risks in Dongting Lake floodplain, China"

_EGUsphere, 2025_

## Author Response (AR1)

Dear Dr. Susanne Liebner,

Thank you very much for giving us the opportunity to revise our manuscript entitled "Vegetation-mediated surface soil organic carbon formation and potential carbon loss risks in Dongting Lake floodplain, China". We appreciate your positive assessment and are encouraged that you and the reviewers acknowledge the quality and merit of our study.

We have carefully considered all the comments from the two anonymous reviewers as well as the additional points you raised. These comments are very valuable and have helped us significantly improve the clarity and rigour of our work.

In addition to addressing all the reviewers' points, we have also provided detailed explanations for the two specific issues you mentioned.

We have read all the comments carefully, responded to them point-by-point below, and revised the manuscript accordingly. In the revised manuscript, new and corrected contents (including references) are marked in red. We hope our revisions have made the manuscript more worthy of publication in Biogeosciences.

Our detailed responses to the comments are as follows:

Comments from the Editor:

Comment 1: Please explain why you used a 0.147 mm sieve and not a 2 mm one commonly used and what potential bias this may have caused.

Response: We thank the editor for this pertinent question. The use of the 0.147 mm sieve was a deliberate choice tailored to our specific analyses. This fine grinding is critical for two reasons:

For Soil $\delta^{13}C$ and $\delta^{15}N$ Isotopes: The analysis requires extremely small (milligram-scale) subsamples. Without fine homogenization, the measured isotopic values could be skewed by microscopic heterogeneity.

For Total TN and SOC: Beyond reducing subsampling error, fine grinding creates a more reactive surface area, ensuring complete combustion and decomposition of the sample during the high-temperature analytical process, which is essential for obtaining accurate and reproducible results.

Our procedure was as follows: the air-dried soil was first passed through a 2 mm sieve to obtain the fine earth fraction. The sieved soil was thoroughly mixed and divided into two portions by quartering. One portion was reserved for other measurements, while the other was finely ground to pass through a 0.147 mm sieve for the aforementioned analyses. This protocol is standard for such measurements and is not expected to alter the intrinsic chemical composition of the soil. Therefore, we are confident it introduces no systematic bias, and we have clarified this in the Methods section (Lines 143-146).

Comment 2: Also, give some explanation on the application of a gas-chromatography-isotope ratio mass spectrometer to measure solid matter (lines 169/170).

Response: We thank the editor for raising this point regarding instrument description. Upon careful verification, we acknowledge that the term "gas chromatography-isotope ratio mass spectrometer (GC-IRMS)" in the original manuscript was imprecise for characterizing bulk solid samples.

The Thermo Scientific Delta V Advantage is an integrated platform capable of hosting different inlet systems. For the measurement of bulk soil $\delta^{13}C$ and $\delta^{15}N$, the analysis was performed using

the Element Analyses-Isotope Ratio Mass Spectrometry (EA-IRMS), which is the standard and most appropriate method for this purpose.

We sincerely apologize for the confusion caused by this oversight. The Methods section has been revised to accurately describe the technique as EA-IRMS. Please see L181 in the revised manuscript.

Response to Reviewer #1

1. L25: Refine "surface SOC".

Response: Thanks for the suggestions! L25 has been changed to: characterize surface SOC (0-20 cm) composition across three dominant vegetation communities. Please see L25 in the revised manuscript.

2.L29: Ensure the unit is consistent with that used in L28.

Response: Thanks for the suggestions! We have adjusted the unit to be consistent with the previous one. Please see L29 in the revised manuscript.

3. L33: Please write out the full name of "POM" upon first mention; the description of spatial heterogeneity in POM contributions is unclear.

Response: Thanks for the suggestions! We have stated the abbreviation (POM) at the location of the first particulate organic matter appearance; We have revised the description of spatial heterogeneity in POM contributions to "the spatial heterogeneity of the POM contribution to the surface SOC". Please see L33-34 in the revised manuscript.

4.L37: Key values should be provided.

Response: Thank you very much for your detailed suggestion. L37 has been changed to: *Miscanthus* soils exhibited enhanced O-alkyl C content (44.75 %) (Alip/Arom:3.64) and reduced aromaticity (0.22) /hydrophobicity (0.68) indices. Please see L38-39 in the revised manuscript.

5.L57-66: Repeated organizational language with unclear expression; not supported by available literature.

Response: Thanks for the suggestions! L57-66 has been changed to: The sources of SOC vary significantly among different vegetation communities, depending on vegetation characteristics and hydrological conditions (Ni et al., 2025; Guo et al., 2025). For instance, in mangrove ecosystems, SOC is primarily derived from mangrove plant tissues, whereas in adjacent S. alterniflora marshes and tidal flats, it relies more heavily on fluvially imported particulate organic matter (POM) (Wang et al., 2024a). Vegetation influences SOC sources mainly through plant productivity and litter decomposition rates, while hydrological conditions regulate the input and deposition of allochthonous carbon (Guo et al., 2025; Xia et al., 2021). Moreover, even within the same type of vegetation community, SOC sources may exhibit spatial heterogeneity due to local topographic features and anthropogenic activities, leading to the accumulation of allochthonous carbon (Swinnen et al., 2020). Please see L58-69 in the revised manuscript.

6.L102: Change "SOC levels" to "SOC content".

Response: Thanks for the suggestions! We have changed "SOC levels" to "SOC content". Please see L101 in the revised manuscript.

7. L221: Provide the calculation formula for A/O-A.

Response: Thanks for the suggestions! We have provided the calculation formula for A/O-A, A/O-A= alkyl C/ O-alkyl C. Please see L237 in the revised manuscript.

8.L256-257: Please verify the data: "The annual sediment transport of the four tributaries was 484.1 × 10⁴ tons".

Response: Thank you for remind us. We have verified that this data is correct.

9.L267: Do the data in Table 1 represent mean ± standard deviation or standard error? Please clarify.

Response: Sorry we did not define the presentation format of the data. The expressed data represents mean ± standard error, which we have added in Table 1. Please see L285 in the revised manuscript.

10.What does the asterisk (*) in the figures represent?

Response: Sorry we did not specify what "*" represents. * indicates significant differences between different vegetation types at the $P<0.05$ level. We have already provided explanations in the caption. Please see L316-317 in the revised manuscript.

11.Discussion 4.1: suggested comparison of soil organic carbon content with other wetlands.

Response: Thanks for the suggestions! We added comparisons with soil organic carbon content of other wetlands, specifically "The surface SOC content of the Dongting floodplain wetland (11.12 g kg$^{-1}$) was close to that of the Poyang Lake wetland (9.69 g kg$^{-1}$) (Yuan et al, 2023) but lower than that of the forested wetland in the middle and lower Elbe River in Germany (33.73 g kg$^{-1}$) (Heger et al, 2021)." Please see L385-388 in the revised manuscript.

12.L420-422: There is an error in the way this sentence is expressed. O-alkyl C constitutes the dominant fraction (27.3–46.8 %) of SOC in Dongting Lake wetlands Consistent with other research findings, rather than the cellulose and hemicellulose components of plant litter decompose rapidly to produce carbohydrates.

Response: We are sorry for the mistake. L420-422 has been changed to: Specifically, the cellulose and hemicellulose components of plant litter decompose rapidly to produce carbohydrates (Mckee et al., 2016). O-alkyl C has also been found to be the dominant fraction of SOC in other lakes or river wetlands (Yang et al., 2023; Wang et al., 2011). Please see L441-442 in the revised manuscript.

13.L449-456: The risk of loss of soil carbon pools in Miscanthus community is higher......, this is an important conclusion. I suggest you add some scientific advises to migrate this loss potential at the end of this paragraph.

Response: Thanks for the suggestions! We have added some scientific advises to migrate this loss

potential at the end of this paragraph, specifically"Therefore, hydrological management strategies such as regulating water levels or extending flood duration could be applied to maintain anaerobic conditions in *Miscanthus* soil, thereby potentially reducing the decomposition rate and loss of SOC." Please see L477-480 in the revised manuscript.

Response to Reviewer #2

1.L89, "sources and stability": The authors infer stability from SOC structure, which they analysed using the 13C NMR method - is this unambiguously possible?

Response: We thank the reviewer for raising this important point regarding the multifaceted nature of SOC stability. We fully agree that stability is not determined by chemical structure alone. In addition to chemical structure, physical (e.g., aggregation) and mineral protection mechanisms also play important roles in SOC stability, which are not directly captured by our $^{13}$C NMR data. However, $^{13}$C NMR is a well-established method for characterizing the chemical composition of SOC and assessing its intrinsic chemical stability (Helfrich et al., 2006;Wang et al., 2025). The ratio of recalcitrant to labile SOC compounds serves as a key indicator of SOC decomposability (Ji et al., 2020). Therefore, the chemical aspect of stability, which our data directly addresses, remains a valuable and established indicator of SOC decomposability. To address the reviewer's concern, in the revised manuscript, we will add a sentence in the relevant section to clarify that the inferred stability is based on chemical structure and acknowledge the potential influence of unmeasured physical and mineral protection mechanisms. Please see L480-483 in the revised manuscript.

2.L98ff: The hypotheses should be briefly justified not just stated

Response: Thank you for your valuable comments. We have outlined the rationale for the hypotheses. The hypotheses of this study were as follows: (1) Regarding vegetation communities, SOC content was expected to be highest in the *Miscanthus* community, intermediate in the *Carex* community, and lowest in the mudflat. This was based on the corresponding gradient in plant biomass input. Spatially, a gradient of East > South > West Dongting Lake was anticipated, owing to the longer inundation durations in East Dongting Lake, which promote anaerobic conditions that suppress SOC decomposition. (2) SOC in the *Miscanthus* and *Carex* communities would be primarily originate from autochthonous plant sources, driven by in-situ plant litter deposition. In contrast, SOC in the Mudflat would primarily originate from allochthonous, derived from particulate organic matter delivered by hydrological processes due to the lack of local vegetation. (3) Due to differences in SOC sources, the SOC structure in the *Miscanthus* and *Carex* communities was hypothesized to be dominated by O-alkyl C (reflecting plant-derived carbohydrates like cellulose). Conversely, the SOC in the Mudflat was expected to be richer in aromatic C, as allochthonous organic matter often contains more recalcitrant components. Please see L100-114 in the revised manuscript.

3.L131: 31 sampling sites of 1x1 m for area of 2564 km² - please explain why these sites were

representative of the entire floodplain.

Response: We appreciate the reviewer's comment on the representativeness of our sampling design. Our sampling strategy was designed explicitly to capture the environmental drivers of SOC sources and stability across the floodplain. The 31 sites systematically cover the key gradients in hydrology (between the three sub-lakes) and vegetation (*Miscanthus*, *Carex*, Mudflat), which are the dominant factors controlling allochthonous vs. autochthonous carbon inputs and the biochemical stability of SOC. Therefore, this stratified design ensures the ecological representativeness of our sampling strategy by capturing the core biogeochemical gradients relevant to our research on SOC sources and stability.

4.L149: units of VOC are inconsistent with eq. 1

Response: Thank you for pointing out the inconsistency in units. We have revised the unit of VOC from g kg⁻¹ to t C / t biomass to ensure dimensional consistency in Equation 1. Please see L154 in the revised manuscript.

5.L161: sum ID is unclear if I_WD is already the number of days when WD>0

Response: Thank you for your valuable comments. The reviewer is correct that the original formula and textual description were poorly presented and created confusion. We deeply regret this error. However, we wish to clarify that the actual calculation performed in our analysis was correct. In practice, we programmatically counted the number of days in the year when the daily water depth (WD) exceeded zero. The formula was an incorrect representation of the underlying analysis. We have revised the manuscript to address this issue thoroughly. The corrected text and formulae are as follows:

"The inundation duration (ID) for each site was calculated as the total number of days within a year when the daily water depth (WD) was greater than zero. This was computed using a daily indicator function, summed over the entire year:

$$ID=\sum_{i=1}^{n} \mathbf{1}_{\{WD_i>0\}} \tag{2}$$

where n is the total number of days in a year, i is the day index, and $\mathbf{1}_{\{WD_i>0\}}$ is the indicator function which takes the value of 1 if the condition $WD_i>0$ is true on the i-th day, and 0 otherwise. The daily water depth $WD_i$ was computed as:

$$WD_i=WL_i - E \tag{3}$$

where $WL_i$ is the daily water level (m) at the Chenglingji (EDL), Xiaohezui (SDL), and Nanzui (WDL) Hydrological Stations, and E is the elevation (m). Please see L163-174 in the revised manuscript.

6.L221: four indicators are presented as equations without numbering. What are the units of the terms presented in those equations?

Response: Thank you for the comment. The equations for the four indicators have now been numbered in the revised manuscript(L237、L240、L243、L246). These indicators are unitless ratios, as they are calculated from the relative peak areas of specific carbon functional groups obtained from the ¹³C NMR spectra. These peak areas represent the proportion of each functional group relative to the total spectral signal.

7.L238: Please explain more detail on the approach to quantify source contributions.

Response: Thank you for this suggestion. We have now expanded on the methodology for quantifying source contributions. In the Bayesian mixing model, the Markov chain Monte Carlo (MCMC) algorithm was set to "normal". Model convergence was assessed using Gelman-Rubin diagnostics and Geweke diagnostics (Stock and Semmens, 2016). Additionally, an "uninformative" prior was selected, and the error structure was defined as "residual and process error". Please see L202-206 in the revised manuscript.

We have made the following additional revisions to the manuscript:

1.The name of the secondary affiliation has been updated to: National Field Scientific Observation and Research Station of Dongting Lake Wetland Ecosystem in Hunan Province, Changsha 410125, China. Please see L6-7 in the revised manuscript.

2.In response to a query raised during the initial review process, Figure 1 in the manuscript has been revised to remove the labels for territories with disputed status under United Nations classifications.

3. The Acknowledgments section has been updated to correct the funding information, which now reads: " This work was supported by the National Key Research and Development Program of China (2022YFC3204101, 2023YFF0807202), the National Natural Science Foundation of China (U22A20570 and U2444221), the Natural Science Foundation of Hunan Province (2025JJ20039), the Science, Technology and Innovation Platform Plan of Hunan Province, China (2022PT1010), and the Comprehensive Investigation and Potential Evaluation of Natural Resources Carbon Sink in Southern Hilly Region, China (DD20220880)." Please see L510-516 in the revised manuscript.